# Evaluation of the Relationship between Spatio-Temporal Variability of Vegetation Condition Index (VCI), Fire Occurrence and Burnt Area in Mount Kenya Forest Reserve and National Park

Kevin W. Nyongesa [1,2,*], Christoph Pucher [1], Claudio Poletti [3] and Harald Vacik [1]

1 Institute of Silviculture, Department of Forest and Soil Sciences, University of Natural Resources and Life Sciences, Vienna (BOKU), Peter-Jordan-Strasse 82, A-1190 Vienna, Austria; christoph.pucher@boku.ac.at (C.P.); harald.vacik@boku.ac.at (H.V.)

2 Department of Natural Resources, Egerton University, P.O. Box 536-20115 Egerton, Kenya

3 Independent Researcher, 17 Via Saletto, Storo, 38089 Trento, Italy; claudio.poletti60@gmail.com

* Correspondence: kevisson2005@yahoo.com; Tel.: +25-474-6086-035

**Abstract:** Climate change, vegetation dynamics, human activities and forest management influence the occurrence of fires. This study investigated the spatio-temporal variability of the Vegetation Condition Index (VCI) and its influence on fire occurrence in three different land use types in Mount Kenya Forest Reserve and National Park (MKFRNP): National Park (NP), Forest Stations (FS) and Farmlands (FL). The study used MODIS satellite data to obtain the Normalized Difference Vegetation Index (NDVI), the VCI, the number of fires and the burnt area. The specific objectives of this research were (i) to examine the spatio-temporal variability of VCI, fire occurrence and burnt area in MKFRNP from 2003 to 2018 and (ii) to explore the relationship between VCI, fire occurrence and burnt area in different areas of the MKFRNP (NP, FS and FL). The findings show that even though fires occur throughout the year in MKFRNP, most of the fires occur during dry seasons. The relationship between spatio-temporal fire occurrence and VCI distribution is different for each land use type. In the FL, the probability of fire ignition and the number of fires per month was more or less the same irrespective of the VCI because of the traditional use of fire as a land management tool. However, the probability of fire ignition and the number of fires per month is high in the NP and FS when the VCI is below 50% (drought), especially in the dry seasons, when and where the impact of meteorological conditions and climate have much more impact than human activities. In addition to the efforts already made by communities, KFS and KWS in the fire fighting and monitoring system, satellite data can be useful to acquire accurate and timely information on the VCI and the likely spatio-temporal occurrence of fires in order to be prepared in the most fire-prone periods and improve fire management, the planning of resources and fire suppression activities in MKFRNP.

**Keywords:** fire occurrence; spatio-temporal; MODIS; VCI; time series; forest; national park; farmlands; burnt area; number of fires

## 1. Introduction

Humans initiated a new stage of ecosystem fire by using it as a land management tool to manipulate vegetation composition, structure, and fuel loads on farmlands, rangelands and other wildland ecosystems [1]. Human interaction with fire and vegetation has occurred at many levels of human population density and cultural development, from subsistence cultures to highly technological societies [2,3]. As a result, almost every landscape has a complex history of human land use and natural disturbances [2], and the distinction between 'natural' and 'cultural' landscapes is not always obvious [3].

The history of people using fire in Africa goes back hundreds of thousands of years. As a result, Africa has a long history of communities using fire as a land management

tool. Therefore, appreciating the evolving relationships and geographic patterns of anthropogenic landscape burning in Africa is crucial because the survival of many species and ecosystems hinges on understanding the historical range of variability in fire activity [4]. The continuous use of fire in Africa has been culturally framed and transmitted, and it is continuing to undergo rapid changes in expression [4,5].

In Kenya, perennial grassland fires are common in many parts of the country because each year, pastoralist communities usually set grasslands on fire to keep them open and to facilitate the growth of new grass for livestock, especially before the rain begins [5]. The Mount Kenya Forest Reserve and National Park (MKFRNP) located in the Mount Kenya region is highly prone to fires. Investigations of paleoecology in the Mount Kenya region indicate that fires have occurred in the area since at least 26,000 years before present (BP) [6]. A study by [7] mapped burn scars on Mount Kenya using satellite data to reconstruct recent fire history. Fire has influenced the vegetation in the landscape and some plant species found in the MKFRNP require fire to germinate, establish or reproduce, and total fire suppression not only eliminates these species but also affects the animals that depend upon them [8,9].

Climate change, vegetation dynamics, human activities and forest management influence the occurrence and intensity of fires in MKFRNP [5,10]. MKFRNP has experienced several extreme droughts and wildfire events in the past decades. Records of the Kenya Forest Service (KFS) and the Kenya Wildlife Service (KWS) show several types of fire related to the heterogeneity of vegetation composition; indeed, ground fires, surface fires and crown fires have occurred in grasslands, farmlands, plantations, bushlands and forests in MKFRNP [4,6]. Their impacts on vegetation conditions and wildfires in MKFRNP have been of great concern to the KFS, KWS and communities [3]. The fire regimes have been influenced by changes in temperature, rainfall, humidity, wind and the amount of $CO_2$ in the atmosphere [4], and climate change may affect fire season length and severity [5]. The increasing human activities and the intensified cultivation of exotic fire-prone tree species in MKFRNP by the KFS are likely to increase the fire hazard in the future [4].

The Kenya Grass Fire Act, Cap 327, provides regulations for planned burnings of bushes, shrubs, grass, crops and stubble within protected areas. However, total fire suppression policy, which aims to detect and suppress all fires, ignoring seasonal weather conditions and land-use types, is practiced by KFS and KWS and, in combination with other human-caused environmental changes, has resulted in extreme wildfires in MKFRNP [5,6]. Additionally, limited government funds to tackle wildfire issues, the retrenchment of human resources within the KFS and KWS and the lack of adequate equipment and well-trained firefighters have seriously affected the capacity to effectively monitor and combat unwanted wildfires in MKFRNP [4,5]. The current conventional fire monitoring activities by KFS and KWS do not detect fires in near real time and are labor-intensive and expensive to conduct over this expansive landscape with varied land uses, topography, vegetation conditions and weather patterns [5,6,10]. Therefore, there is a need for the KFS, KWS and communities to better understand how to use satellite data to obtain near real-time information on fire ignitions, burnt area, topography, vegetation condition, weather and other parameters considerably at fine spatio-temporal resolutions, as this will help them to improve land-use-specific fire monitoring, wildfire management strategies and habitat-specific firefighting in MKFRNP.

Studies show that satellite missions like MODIS, Landsat and Sentinel enable multi-spectral, thermal, as well as radar data in high spatial and temporal resolutions [11]. The new satellite multi-sensor system enables environmental monitoring in almost daily revisit time globally [12]. Satellite data allows for the timely (near real time) acquisition of information of the factors that can help to reconstruct fire history, locate fire hot spots, map burnt areas, determine the number of fire ignitions, the topography, the vegetation conditions, the weather and other parameters at considerably finer spatio-temporal resolutions [13]. A study by [14] identified burnt areas in Mount Kenya using Landsat 8 multispectral imagery. Another by [15] used Geographic Information System (GIS) to assess the actual and mod-

eled burnt area accuracy, differences in fire spread and behavior through the simulation of recent forest fires that affected the Mount Kenya Forest Reserve. However, so far, no study has evaluated the relationship between the spatio-temporal variability of the vegetation condition, the fire occurrence and the burnt area in the Mount Kenya Forest Reserve and National Park.

Assessing the dynamics of vegetation and its response to environmental changes is essential to better understanding ecosystem changes and the sustainable management of forest resources [3]. The vegetation indicators Normalized Difference Vegetation Index NDVI and Vegetation Condition Index (VCI) provide alternative measurements of the relative vegetation health [11]. Time series of the NDVI and the VCI derived from satellite data have been used to study fires in different parts of the world [16]. A time series of the NDVI can allow one to visualize and quantify vegetation health before and after the fire [16]. However, [16] found that NDVI values in areas where fires took place were similar to NDVI values in areas in which fire did not occur, showing the limitations of using the NDVI as an index of fire occurrence and risk. VCI seems to be more suitable for fire monitoring because it reduces the impact of ecosystem-specific responses (e.g., driven by climate, soils, vegetation type and topography) and enhances inter-annual variation [17].

The VCI contains both real-time and historical information about the NDVI [11,18], which allows the monitoring of vegetation that may be stressed due to potential drought [19] and may be fire-prone as well. Very low VCI values (close to zero percent) reflect an extremely dry month, VCI values around 50% reflect fair conditions and VCI values above 50% indicate optimal conditions [20,21]. However, the relationship between drought and fire is complex, as the timing, intensity and frequency of drought events have divergent impacts on fuel flammability and fire behavior [2]. This seems to be highly relevant in the context of a changing climate, as there are indications that seasonal alterations and more inter-annual variability might influence the fire regime [5].

This paper evaluates, therefore, the relationship between the spatio-temporal variability of VCI, fire occurrence and burnt area in three land use types: National Park (NP), Forest Stations (FS) and Farmlands (FL) in MKFRNP. The main hypothesis is that VCI has an effect on the number of fires and amount of burnt area in the NP, FS and FL in MKFRNP. The two specific objectives of this publication are (i) to examine the spatio-temporal variability of VCI, fire occurrence and burnt area in NP, FS and FL in MKFRNP from 2003 to 2018 and (ii) to explore the relationship between VCI, fire occurrence and burnt area in NP, FS and FL in MKFRNP from 2003 to 2018.

## 2. Materials and Methods

### 2.1. Description of the Study Site

2.1.1. Management of Mount Kenya Forest Reserve and National Park

MKFRNP is a UNESCO World Heritage Site. It is located to the east of the Great Rift Valley, along latitude 0°10′ S and longitude 37°20′ E. Mount Kenya was formed as a result of volcanic activity with a base diameter of approximately 120 km [6,8] and has an original elevation of more than 6000 m asl. The MKFRNP has two main administrative areas. Mount Kenya Forest Reserve is managed by the KFS and covers 213,083 ha. It has been divided into 19 well-defined FS that are management unit areas that have well-defined administrative boundaries. The NP covers 69,406 ha and is managed by the KWS. The Mount Kenya Forest Reserve management plan (2010–2020) and the Mount Kenya Ecosystem management plan (2010–2020) stipulate rules for the establishment of these management zones and guide the management and future planning to avoid conflicts among different users [6,8]. The zoning of MKFRNP into FS, NP and FL has an influence on human activities allowed in those zones, which affect the ignition probability of fires accordingly [5,6,8]. The whole study area of this analysis represents an area of around 9240 km$^2$ (Figure 1).

2.1.2. Climate of Mount Kenya Forest Reserve and National Park

The climate of MKFRNP is largely determined by altitude. There are large differences in altitude within short distances, which determine a great variation in climate over relatively small distances. Average temperatures decrease by 0.6 °C for each 100 m increase in altitude. At the base, mean temperatures are around 20 °C, whereas at the top, temperatures are below 0 °C. Rainfall is moderate on the lower slopes and heavier higher up [8]. The altitudes with the highest rainfall are between 2700 and 3100 m, while above 4500 m, most precipitation falls as snow or hail. Rainfall pattern in MKFRNP ecosystem is bimodal. It ranges from 900 mm in the north (leeward side) to 2300 mm on the southeastern slopes (windward side) of the mountain. The wet seasons are from April to June and from October to December when it is drizzly and cloudy, and the dry seasons are from July to September and from January to March [8]. To illustrate precipitation patterns in the MKFRNP we used the Climate Hazards Group Infrared Precipitation with Stations (CHIRPS) v2.0 data set. CHIRPS is a quasi-global monthly rainfall data set at 0.05° spatial resolution [22]. We downloaded rainfall data of study area from CHIRPS from 2003 to 2018 (Table 1). We calculated the monthly pixel time series of precipitation in study area derived from the CHIRPS data and obtained the spatial pattern of precipitation in MKFRNP (Figure 1).

2.1.3. Vegetation Types in Mount Kenya Forest Reserve and National Park

Vegetation types and species distribution in MKFRNP follow different elevation and climatic zones [6,8]. There are six major vegetation zones that have been classified according to altitude and floristic composition. They are montane forest 1600–2400 m; bamboo thickets 2400–2850 m; Hagenia-Hypericum woodland 2850–3000 m; Erica bushland/shrubland 3000–3300 m; alpine zone 3300–4350 m and nival zone 4350–5199 m [23]. The FS has heterogeneous vegetation that is characterized by indigenous and exotic tree species, bamboo, shrubs and grasslands (Figure 1). The bamboo (*Arudinaria* spp Michaux) zone occurs between 2500 and 3200 m asl, but it is absent on the northern side of MKFRNP due to drier conditions [8]. The indigenous forest zone starts at 2400 m down to 3000 m asl and is dominated by yellowwood (*Podocarpus latifolius* Thunberg Ex Mirbel) mixed with brittle-wood (*Nuxia congesta* R. Brown Ex Fresen.) at the upper altitudes [6,8], while moist forests of East African camphorwood (*Ocotea usambarensis* Engler), forest newtonia (*Newtonia buchananii* Buchanan) and woodland croton (*Croton sylvaticus* Hochst. Ex C. Krauss) occur at lower altitudes between 1450 and 2400 m asl [6,9]. There are plantation forests of exotic tree species that were established by KFS in MKFRNP and are prone to crown fires like cypress (*Cupressus lusitanica* Mill.), patula pines (*Pinus patula* Schiede Ex Schltdl. and Cham), radiata pines (*Pinus radiata* D. Don), blue gum (*Eucalyptus saligna* Smith) and rose gum (*Eucalyptus grandis* W. Hill Ex Maiden). Plantations of indigenous species like Meru oak (*Vitex keniensis* Turrill) and African pencil cedar (*Juniperus procera* Hochstetter. ex Endlicher) were also established by the KFS in MKFRNP [6,8]. The moorland in the NP area is characterized by homogenous Ericaceous vegetation. The Ericaceous zone lies between 3000 m and 3500 m asl and is mainly covered with giant heath, African sage (*Artemisia afra Jacquin Ex* Willdenow), several gentians (*Swertia* spp Linnaeus), smaller trees in glades, such as the East African rosewood (*Hagenia abyssinica* Willdenow), St. John's wort (*Hypericum* spp. Linnaeus) and trees that are covered with moss and lichens (*Usnea* spp. *Dillenius Ex* Adanson) [6,8]. Farmlands in and around MKFRNP are usually characterized by perennial and annual crops [6–9] (Figure 1).

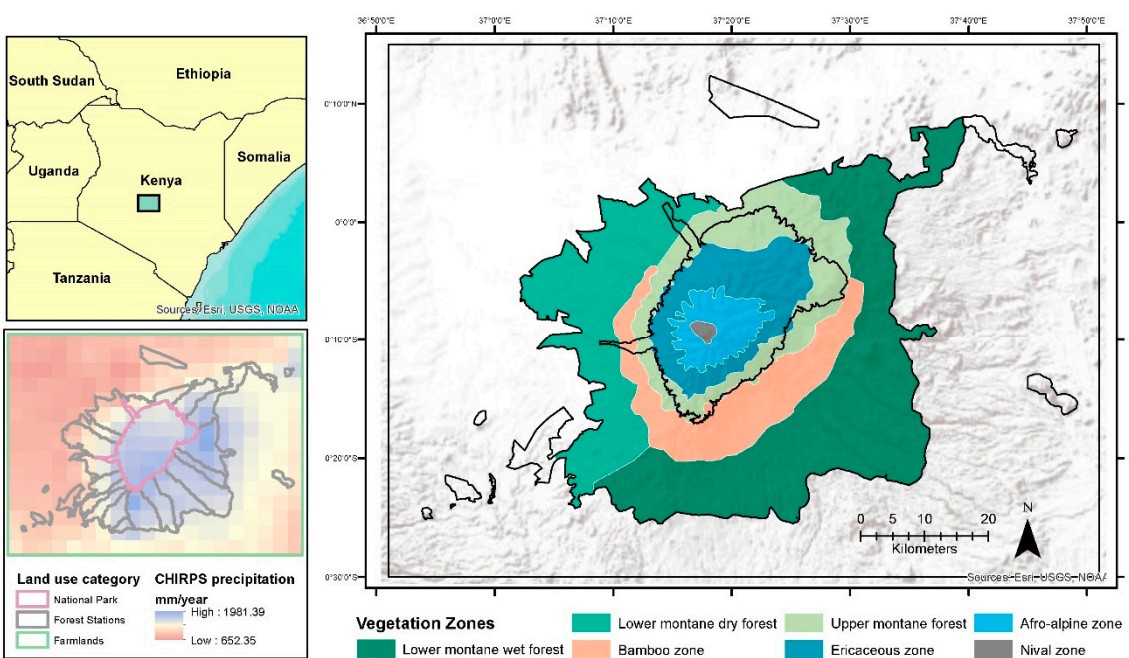

**Figure 1.** Location of Mount Kenya Forest Reserve and National Park (Top left), the spatial pattern of precipitation in Mount Kenya Forest Reserve, National Park and Farmlands from 2003 to 2018 (Source: Climate Hazards Group Infrared Precipitation with Stations (CHIRPS) (Bottom left), and the vegetation zones map of Mount Kenya Forest Reserve and National Park (adapted from [23,24] (Right).

### 2.2. NDVI, VCI, Fire Occurrence and Burnt Area Data

We employed a filtered 7-day NDVI time series with a spatial resolution of about 250 m spanning from 2003 to 2018 from BOKU's MODIS data processing chain (Table 1) [25]. The data are operationally used by Kenya's National Drought Management Authority (NDMA) and are available at (https://ivfl-arc.boku.ac.at/kenya/map/ accessed on 18 October 2020). The latest version of the processing chain relies on the MOD09Q1 and MYD09Q1 Collection 6 surface reflectance products, which are obtained from NASA's LP DAAC [26]. Normalized Difference Vegetation Index (NDVI) images were calculated with 8 days of temporal and 250 m spatial resolution for each pixel. The NDVI data were filtered with the Whittaker smoother in near real-time, which fits a discrete series to discrete data and puts a penalty on the roughness of the smooth curve [27]. The smoothing of the NDVI data takes the quality of the observations according to the MODIS VI Quality Assessment Science Data Set (QA SDS) and the compositing day for each pixel into account [25]. The details of the processing chain were presented in [25]. Monthly NDVI values were calculated by averaging the 7-day images for each pixel.

We calculated monthly VCI by applying the following Equation (1) on the final NDVI data for each month on each pixel.

$$VCI = 100 \times \frac{NDVI_i - NDVI_{min}}{NDVI_{max} - NDVI_{min}} \tag{1}$$

An illustration of the calculation of (1) is given for the month February 2006: NDVIi would be the monthly NDVI values by averaging the 7-day images for each pixel for February 2006. NDVImin is the minimum NDVI value observed for February from 2003 to 2018 for that pixel. NDVImax is the maximum NDVI value observed for February from 2003 to 2018 for that pixel. The numerator is the difference between the actual and the minimum values of the NDVI and is indicative of the meteorology and vegetation information of a specific period.

Daily fire occurrence data ranging from January 2003 to December 2018 were obtained from the Fire Information for Resource Management System (FIRMS) Archiving and Distributing MODIS Active Fire Data (Table 1) [28]. The MODIS Active Fire Data included information about the confidence of the detected fire. Only fires with a confidence $\geq 30\%$ were included in the analysis. A total of 1300 fires with confidence $\geq 30\%$ were registered in our study area for the time period 2003 to 2018.

Burnt area fire product MCD64A1 was obtained from NASA's Land Processes Distributed Active Archive Center (LP DAAC) [26]. The MCD64A1 from the LP DAAC product combines data from two satellites (AQUA and TERRA) and returns estimated burnt areas monthly based, with a spatial resolution of around 500 m. In this study, the LP DAAC MCD64A1 raster dataset with a spatial resolution of around 500 m was utilized to select burnt areas within the study area (Table 1). Another MODIS Active Fire Data product, MCD14DL-Collection 6, that detected fire occurrences on a monthly basis, was also used in our study [28]. However, the fact that the spatial resolution of the LP DAAC MCD64A1 for detecting burnt areas is about 500 m and permits the detection of burnt areas with a size bigger than 24 ha should be considered, and those smaller than 24 ha may have been left out of the analysis [29]. Possible ignition spots in MKFRNP were displayed to visualize the spatial extent of the burnt areas and ignition sites in the three major land use types: NP, FS and FL.

*2.3. Analysis of the Relationship between VCI, Fire Occurrence and Burnt Area in Mount Kenya Forest Reserve and National Park*

For a detailed analysis of the relationship between VCI, fire occurrence and burnt area the study area was divided into 15 times 20 cells with 0.05° resolution (~5.55 km), following the resolution of the CHIRPS precipitation data [22]. For each analysis cell, 192 monthly values covering the 16-year period between 2003 and 2018 of VCI, number of fires and burnt areas are obtained. Four seasons were distinguished for the analysis: (i) dry season from January to March, (ii) wet season from April to June, (iii) dry season from July to September and (iv) wet season from October to December. Monthly values were temporally aggregated to the four seasons by calculating the mean of the VCI and the sum of the number of fires and burnt area. Each analysis cell is further assigned to one of the three major land use types: NP, FS and FL.

Time series of monthly precipitation and VCI were produced by averaging the monthly values for cells belonging to the same land use category. Time series of seasonal number of fires and burnt areas were produced by summing up the monthly values for cells belonging to the same land use category and season.

Analysis of the spatio-temporal variability of the VCI showed that the average monthly VCI can vary in space, time and within the same or different land use types, as indicated by previous studies [21,25]. Therefore, the relationship between VCI, number of fires and burnt areas was analyzed at raster cell level rather than for a whole land use type in the month of February for each year from 2003 to 2018 because it is the most fire-prone month.

For the analysis on the cell level, VCI values were divided into 10 classes ([0,10], (10,20], (20,30] . . . (90,100]). For each recorded fire and burnt area, the VCI condition in its associated cell at the time of fire occurrence was known. The number of fires and burnt areas were then summed up for each land use category, season and VCI class.

The relationship between VCI and number of fires and burnt area was analyzed using Poisson regression. Poisson regression is suitable for analyzing count data and avoids predicting negative values (e.g., negative number of fires or negative burnt area) often present when using linear regression [30].

Statistical analysis was performed using the statistical software R [31]. Visualization was done using R and ESRI® ArcGIS 10.2.1 [32].

## 3. Results

### 3.1. Spatio-Temporal Analysis of Fire Occurrence, Burnt Area and VCI in Mount Kenya Forest Reserve and National Park

To get a better understanding of the occurrence of fires, burnt area and the variability of VCI in the study area we first analyzed their spatio-temporal pattern. According to MODIS fire data, 1300 fires occurred in the study area during the observation period (Figure 2). A total of 637 fires were recorded in Farmlands (FL), 359 fires in the National Park (NP) and 304 fires in the Forest Stations (FS). Over half of the fires occurred during the dry season from January to March, and another fourth of the fires occurred during the second dry season from July to September. The NP fires occurred in the whole area during the first dry season, while in the second dry season, they were almost exclusively limited to the north-eastern region (Figure 2). During the wet season, fires occurred in the NP in the north-eastern, north-western and south-western regions even if the number of fire events was noticeably low. FS fires were mainly concentrated in the north-west to north and south-east during the first dry season and, similarly to the NP fires, were concentrated in the north-eastern region during the second dry season. The FL fires predominantly occurred in the north-west to northern region during the first dry season and in the west to south-west and eastern region during the second dry season. FL fires were also quite common during the wet season from October to December.

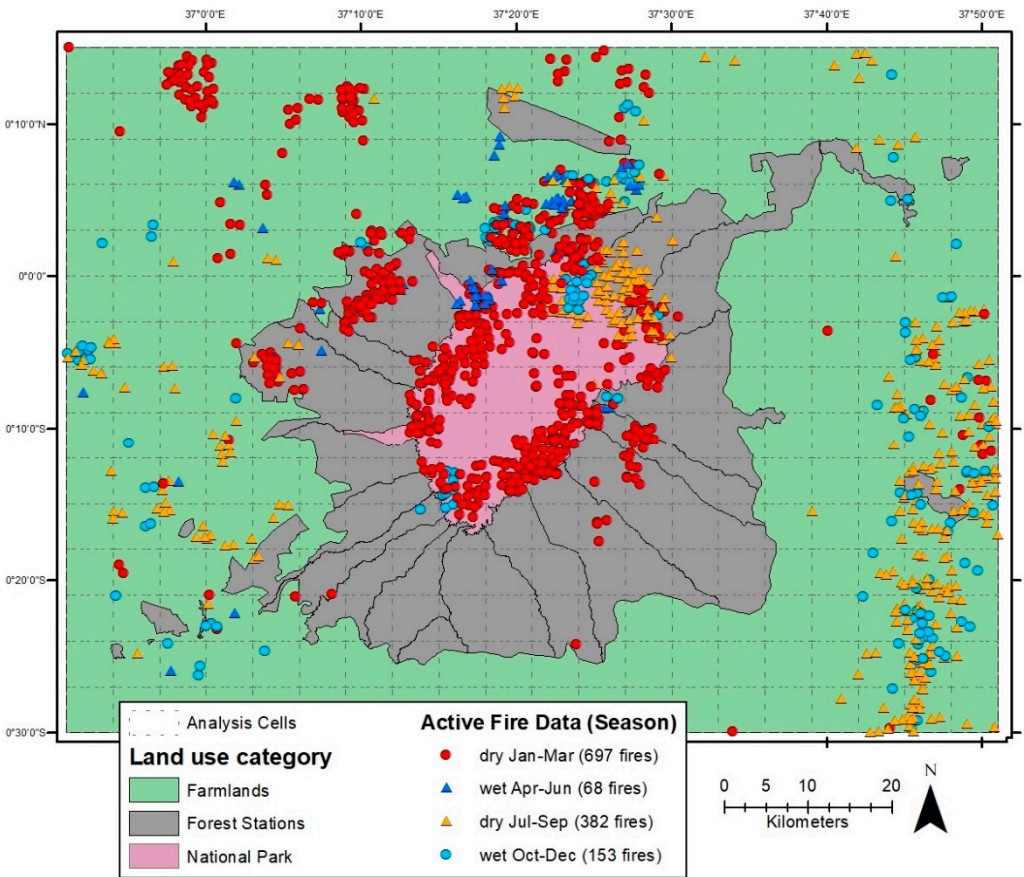

**Figure 2.** Land use types (National Park, Forest Stations and Farmlands) and the spatio-temporal occurrence of fires in Mount Kenya Forest and National Park from 2003 to 2019.

In general, 207 fires were recorded in the MKFRNP in the year 2012, which makes it the year with the most recorded fires. One-fourth and one-third of these fires were recorded in FS and NP, respectively (Figure 3). On the other hand, in the year 2018, the lowest number of fires (17) was recorded. It was found that before the year 2012, around 90 fires occurred per year. However, after the year 2012, around 40 fires occurred per year. In general, years

with a high number of fires also recorded a high amount of burnt area. However, this is not always the case, as can be observed for the years 2003 and 2004, where only a little burnt area was recorded, although several fires occurred.

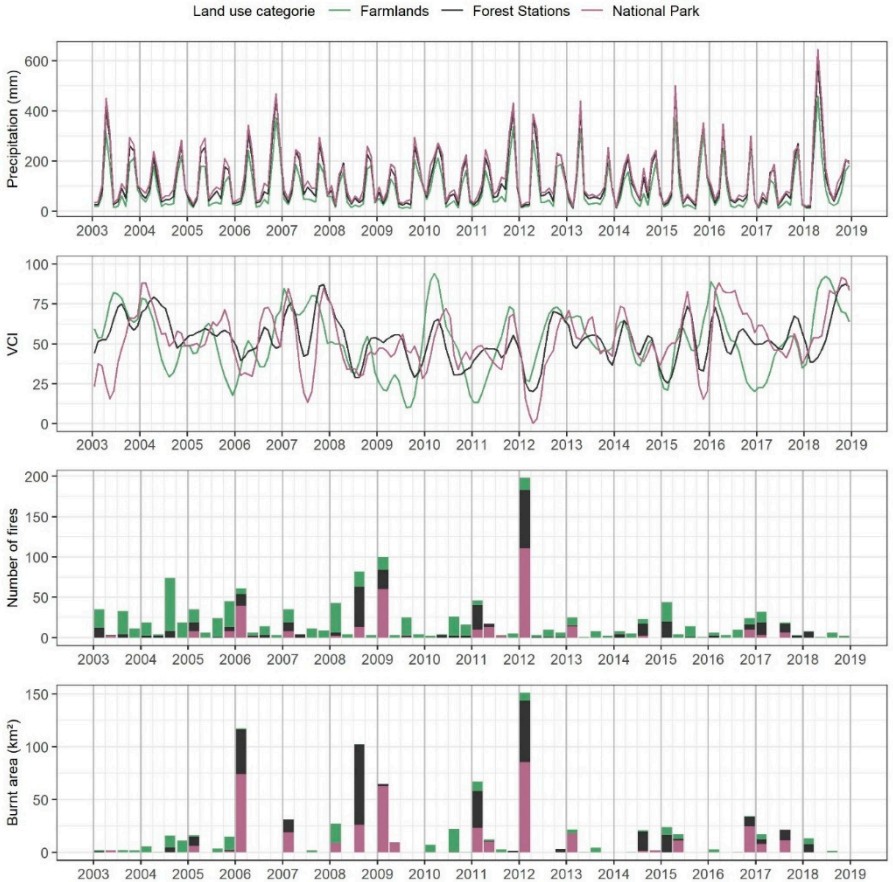

**Figure 3.** Time series of monthly precipitation, vegetation condition index (VCI) and seasonal (quartal) number of fires and burnt area.

The bimodal precipitation pattern is clearly present in the CHIRPS precipitation data and applies to all land use categories with the mean precipitation for most parts also being similar (Figure 3). The VCI shows a more complex pattern with at times diverging behavior between land use categories. For instance, during the first wet season in 2003, a drop in VCI can be observed for the NP, which is not present in the FL and the FS. Sometimes, a lag can be observed, such as during the second dry season in 2005 where the VCI in the FL is already dropping, while it is still rising in the NP and only starts to drop during the second wet season. A similar behavior can be observed in 2016, when the VCI in the FL and FS started to drop earlier than in the NP. During this period, the drop in VCI in the FS was lower compared to the other land use categories and stopped during the first wet season. This also shows that the magnitude of the rise or drop can differ, as can also be observed in 2007, 2010 or 2012, for instance.

In general, the VCI is related to precipitation patterns. High VCI values during the first dry season at the beginning of the year often coincide with higher-than-usual precipitation at the turn of the year (Figure 3). This can be observed in the years 2004, 2007, 2010 and 2016. The year 2018 was an exceptionally wet year, and the consequences are easily visible in the high VCI values. However, at the beginning of the second wet season in the year 2011, the precipitation was high, but less precipitation than usual occurred later in the season and at the beginning of 2012, and there was also an evident drop in VCI (Figure 3). Generally, a distinct drop in the VCI values can be observed in all land use types towards the end of November, and it reached its lowest point during March and April 2012. This

was also the period with the highest number of fires and burnt areas recorded during the observation period (Figure 3).

The VCI not only differs between land use types but also shows complex patterns within the same category (Figure 4). For instance, in February 2013, the eastern part of the FS experienced bad conditions for the vegetation, as shown by the low VCI values, while in the western part, the conditions were still good, as shown by the high VCI values. In 2003, the northern to north-western parts of FL and FS experienced better conditions, as shown by the high VCI values compared to the eastern to south-eastern parts, which had low VCI values (Figure 4).

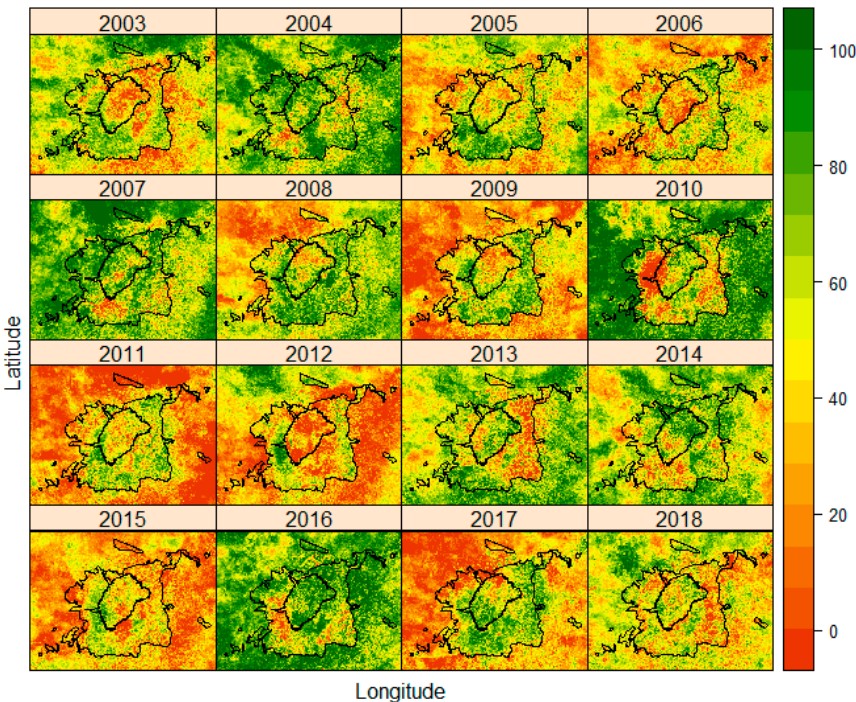

**Figure 4.** Spatio-temporal variability of the VCI in the month of February (National Park, Forest Stations and Farmlands) in Mount Kenya Forest Reserve and National Park from 2003 to 2018.

*3.2. Detailed Analysis of the Relationship between VCI, Fire Occurrence and Burnt Areas at Raster Level*

The Poisson regression allowed us to analyze how the vegetation condition influences the occurrence of fires and the burnt area (Figures 5 and 6). While it can be observed that in the FL, higher VCI classes led to a lower number of fires for both dry seasons, this effect is only weak (Figure 5A). On average, 1.85 fires per year occur during the first dry season, and around 2.4 fires per year occur during the second dry season when the VCI is between 0 and 10, with the number of fires only decreasing by around 6% and 7% per VCI class, respectively. It can also be observed that more fires occur for VCI values between 40 and 70 compared to lower values meaning that a general declining trend is not present. Such a general declining trend between the number of fires and VCI classes can be observed during the second wet season from October to December in FL, with the number of fires declining by 15 % per VCI class. In FL, the VCI has no effect on the burnt area during the dry seasons and the first wet season, but the burnt area decreased by almost 38% per VCI class during the second wet season (Figure 6A).

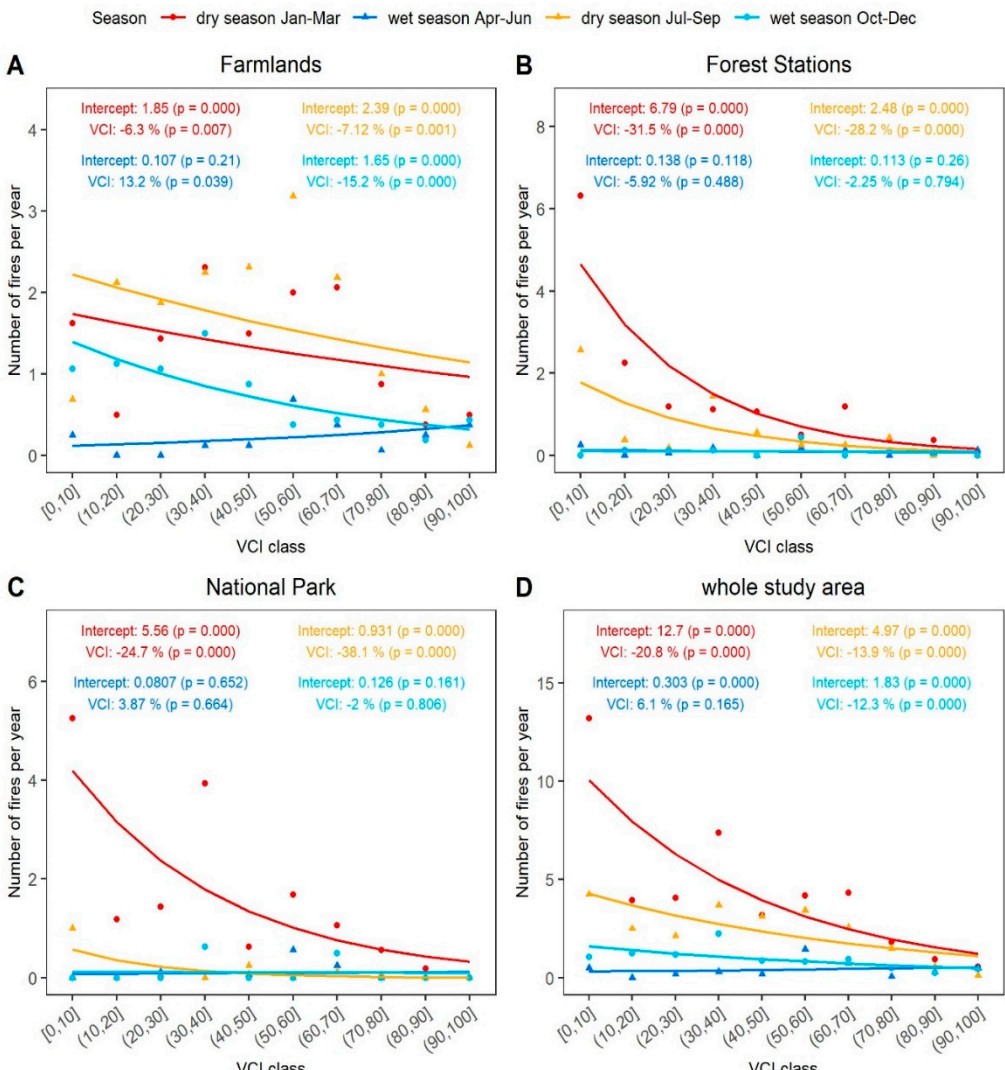

**Figure 5.** Relationship between vegetation condition and number of fires in each of the land use categories (**A**–**C**) and the whole study area (**D**). The vegetation condition is expressed by VCI classes. Each point represents the average number of fires per year recorded in the respective VCI class.

Looking at the FS, a strong trend of reduced occurrence of fires with higher VCI classes can be observed during both dry seasons (Figure 6B). Here, the number of fires decreased by 31.5% in the first dry season and 28% in the second dry season per increase in VCI class. During the wet seasons, the VCI had no effect on the occurrence of fires in FS. The same trends as the number of fires can be observed for the burnt area (Figure 6B).

When looking at the NP, a declining trend in the number of fires with higher VCI classes can be observed for the first dry season from January to March (Figure 5C). There is also a declining trend in the second dry season from July to September. However, there was overall a smaller number of fires during this season compared to the first dry season. The VCI had no effect on the number of fires in NP during the wet seasons. Similar trends can be observed for the burnt area in NP (Figure 6C).

Looking at the whole study area, the effect of the VCI on the number of fires and amount of burnt area was present during both dry as well as the second wet season (Figures 5D and 6D). The effect was highest during the first dry season where, on average, 12.7 fires per year occurred when the VCI was between 0 and 10, and the number of fires decreased by almost 21% per increase in VCI class.

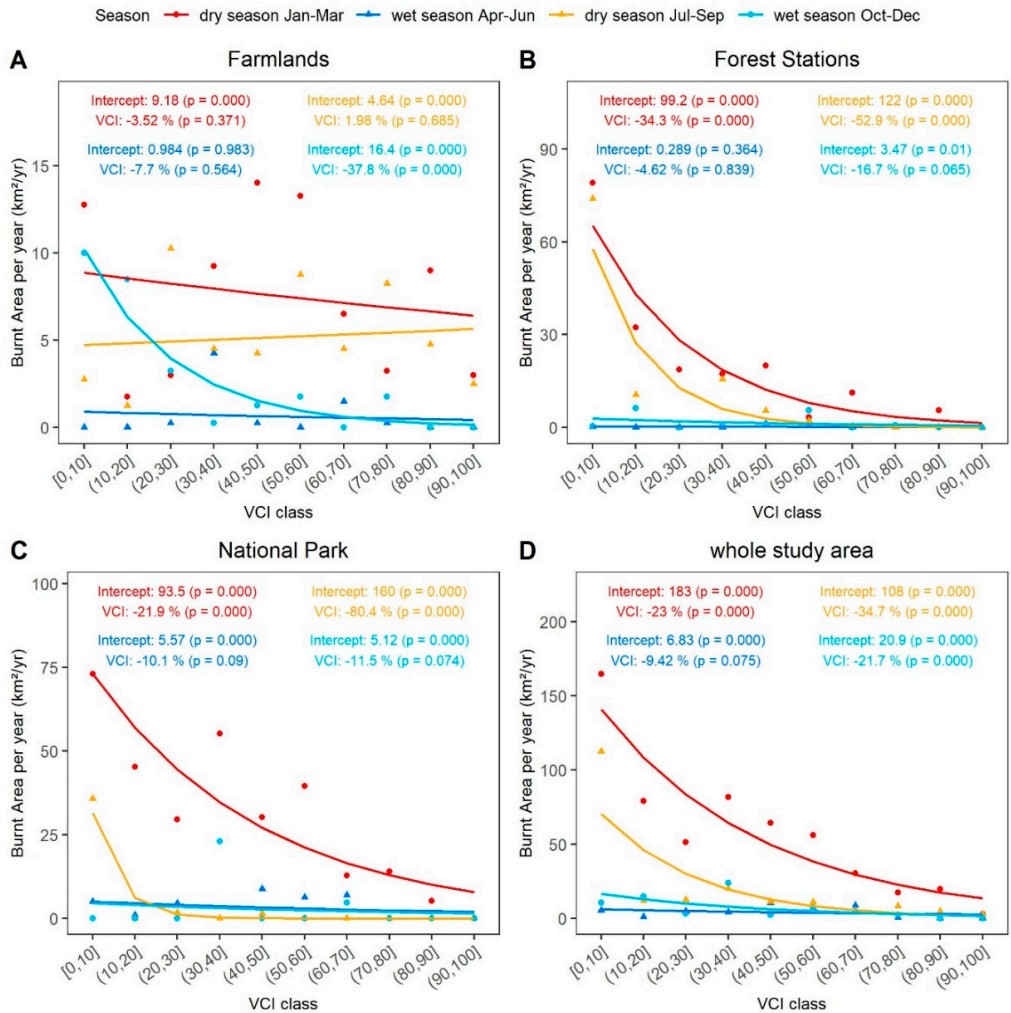

**Figure 6.** Relationship between vegetation condition and burnt area in each of the land use categories (**A–C**) and the whole study area (**D**). The vegetation condition is expressed by Vegetation Condition Index (VCI) classes. Each point represents the average burnt area in square kilometers per year and VCI class recorded during the respective season.

## 4. Discussion

The study investigated the spatio-temporal variability of the VCI and its influence on fire occurrence in three different land use types in the Mount Kenya Forest Reserve and National Park (MKFRNP). The spatial analysis showed that the majority of fires occur in the Farmlands (FL). In the FS, a weak relationship between VCI and number of fires and burnt area was observed even during the dry seasons, with more fires occurring when the VCI was around 50%. These results suggest that fire is indeed an important land management tool for farmers and, consequently, that fire occurrence in FL is more driven by land use activities rather than vegetation conditions. In the National Park (NP) and Forest Stations (FS), a relationship between the VCI and the number of fires and amount of burnt area was observed during the dry seasons. This suggests that in the NP and FS, human-caused fire events are much more related to seasonal weather conditions compared to land use practices.

During the dry seasons from January to March and from July to September, the majority of fires occurred from a temporal perspective. Both the dry conditions leading to a decrease in fuel moisture and accumulation of dead fine fuel [33] as well as seasonal land use activities of local communities contributed to the increased occurrence of fires during these seasons. In the NP and FS, the dry conditions favor outbreaks of unintentional fires,

as community members illegally use fire during the dry season to burn charcoal, harvest wild honey and hunt and roast game meat [4].

Farmers and pastoral communities living around the MKFRNP use fire to prepare farmland or grasslands, especially before the rain begins [6,8,9]. Especially during periods of extreme drought, migrant pastoral communities come to graze their livestock in the NP, set fire to the old grass to facilitate the growth of new grass and then move away in search of good pasture grounds. As such intentional fires due to land management sometimes go out of control, they can contribute to fire outbreaks in the NP and FL and a loss of grazing grounds for the locals who depend on the grasslands within MKFRNP for grazing their livestock. Several plantations of exotic tree species (sp. *Pinus*, sp. *Cupressus*, sp. *Eucalyptus*), which have been established in MKFRNP by the KFS for the pulp and timber industry, additionally contribute to anthropogenic-caused ignitions due to the increased flammability and high fuel loads [4]. The concentration of fires in the north-eastern parts of the NP and eastern parts of Farmlands (FL) during the second dry season from July to September may also be related to a combination of climatic and land use effects. Both areas are located on the windward side of MKFRNP, which receives more rain during the wet season from April to June. The rain is used for the cultivation of several crops like cereals and makes grasses and shrubs grow very rapidly. Grasses, shrubs and cereal crop residues then dry up during the second dry season (July to September), increasing fine fuel accumulation and the continuity of burnable vegetation [6,8]. Especially in farms with cereal residues with high flammability, fires ignite easily and are difficult to extinguish. The land use activities in the FL might also explain the observed relationship between the VCI and fire occurrence during the second wet season from October to December. Many farmers living in and around MKFRNP prefer setting their crop farms and rangelands on fire during the second wet season because the fuels, which include dry bushes, grasses and crop residues, have the optimum fuel moisture content for conducting prescribed burning and the wind speed is not too high to cause a quick fire spread or catastrophic fires [33]. During this wet season, they likely have to wait for drier (lower VCI) conditions to set the fire, which would explain the observed pattern.

The exceptionally high occurrence of fires in 2012 is likely related to the precipitation pattern at the turn of the year 2011. High precipitation at the beginning of the second rainy season in 2011 favored the growth of vegetation. Precipitation then rapidly dropped off at the end of 2011, and almost no rain was falling during the first dry season in 2012. This led to a distinct drop of VCI in all land use categories, as the grasses, shrubs and bushes dried up, leading to an increased fine fuel accumulation, continuity and larger fires in 2012 in MKFRNP. These findings can also be confirmed by [5], who have described how those seasonal alterations and inter-annual variability influences the fire regime. Rapidly drying abundant fuels in forest understories and grasslands after a wet season in MKFRNP may also feed larger fires. However, prolonged drought in MKFRNP may limit fire occurrence, as the availability of fuels (e.g., grasses) is reduced due to lack of precipitation [2].

The decline of fires and burnt area in MKFRNP after 2012 might be related to the increased participation by the community forest associations (CFAs) in fire management activities [6,8]. Once a fire season has been declared by the KFS and KWS managers, fire monitoring is done regularly, and the lighting of fires in the FS and NP is completely forbidden. The KFS, KWS managers and CFA scouts have increased fire monitoring during the fire season, and there has been an increase in the number of people who have been arrested and punished according to the law for lighting fires in FL, FS and NP during the fire season. The KFS and KWS have also been encouraging CFA members, especially farmers living in and around MKFRNP, to create a defensible space, an area around their building/property in which vegetation, debris and other types of combustible fuels are treated, cleared or reduced to slow the spread of fire to and from the buildings or properties [5]. The KFS and KWS have been licensing CFA members to thin and prune trees for poles, collect firewood, cut grass for thatching houses and feed livestock. Recommendations for the prescribed burning indicate that the moisture content of the

dead fuels should be sufficient for the fuels to be burnt and that the moisture content of live fuel should be sufficiently high so that the fire cannot spread into the crown [28,34]. These practices might have helped to reduce fuel loads and minimize the risk of larger fires in MKFRNP.

However, some limitations of this study arose from the inherent use of MODIS satellite data for the number of fires and amount of burnt area. Some MODIS products are based on coarse spatial-resolution images ($\geq$500 m) and may therefore have important omission and commission errors when fires are small in size [9]. The observed discrepancy between the documented number of fires and the amount of burnt area for some seasons, especially in the FL, might be related to the fact that only fires with a confidence $\geq$ 30% were included in this analysis. It must be considered that the confidence percentage of MODIS active fire depends on the temperature values detected. So, a confidence rate of 0% might have occurred in forested areas where the canopy might have blocked the heat signal [29]. This could have led to a lower number of fires in forested areas. Also, the spatial resolution of the LP DAAC MCD64A1 for detecting burnt areas is about 500 m and permits the detection of burnt areas with a size bigger than 24 ha. For this reason, burnt areas smaller than 24 ha might not have been analyzed as well. This finding can be confirmed for the study area by [7], as their results indicate that they were unable to detect smaller fires as well.

## 5. Conclusions

This study found that there is a large spatio-temporal variation in the average monthly VCI values depending on the season, prevailing meteorological conditions, land use type and fires in MKFRNP. Due to climate change and the intensive land use changes, there is a need to consider social, economic, cultural and ecological aspects in minimizing the damage of catastrophic fires and maximizing the benefits of prescribed fires in MKFRNP. Communities living around the MKFRNP have a long history of using fire as a land management tool, but there is a strong need to understand the relationship between VCI, fuel moisture content, ignition probability and fire spread under the changing environmental conditions. Human-caused fire ignitions in FS, NP and FL are more likely to increase in the future because the changing climate may affect fire season length and severity with potentially significant feedback to the Mount Kenya ecosystem [4].

The inclusion of CFA members in the fire fighting system after 2012 ameliorate the fire detection and fire fighting system reducing the number of fires and the dimension of burnt areas. Nevertheless, the current conventional fire monitoring, planning of resources and fire suppression activities in MKFRNP do not detect fires in near real time and are labor intensive and expensive to conduct over expansive landscapes with varied land uses, topography, vegetation conditions and weather patterns. An integrated fire management (IFM) would require a wise combination of early warning systems, prevention measures and suppression techniques that provide accurate and timely information on all fuel types [35]. In this context, the ignitability and combustibility rankings of tree species and the most significant fuel types are essential components in fire prevention and fire risk management [33,36]. Knowledge of fuel conditions constitutes an effective means of preventing or managing fires [37]. Therefore, timely information on VCI needs to be incorporated into fire-related forecasts and long-term fuel management planning to support KFS and KWS managers, communities and fire experts in MKFRNP. There is a need for better integration of the knowledge and resources to enable KFS, KWS and communities to conduct observation and monitoring, prediction and forecasting, communication and outreach to the public. Therefore, the use of satellite data will play a crucial role in obtaining accurate and timely information on the VCI, spatio-temporal occurrence of fires and burnt. The main limitations of this study are related to the use of remote sensing products. As the confidence percentage to MODIS active fire depends on the temperature values detected, and the spatial resolution for detecting burnt area is limited to fires bigger than 24 ha, there is the possibility that a limited number of records can be analyzed. Therefore, further research needs to be done using satellite data obtained at much finer spatial resolutions to

evaluate the importance of human activities, VCI, weather conditions and spatio-temporal occurrence of fire in MKFRNP. Additionally, it would be worth it to analyze longer timer series of fire records in future in order to study the relationship between the VCI, fire occurrence and burnt area in the different vegetation zones of Mount Kenya in addition to the land use categories as well. The fire management activities in MKFRNP should be continued to reduce the fuel loads and create a defensible space in and around properties. In this context, accurate information on fuel moisture is important for scheduling prescribed burning practices by KFS and KWS [33].

**Author Contributions:** The conceptualization was done by K.W.N. and H.V.; methodology, K.W.N., H.V., C.P. (Christoph Pucher) and C.P. (Claudio Poletti); validation, K.W.N. and H.V.; formal analysis, C.P. (Christoph Pucher); investigation, K.W.N., H.V., C.P. (Christoph Pucher) and C.P. (Claudio Poletti); data resources, C.P. (Christoph Pucher); data curation, C.P. (Christoph Pucher) and C.P. (Claudio Poletti); writing—original draft preparation, K.W.N. and H.V.; writing—review and editing, K.W.N., H.V., C.P. (Christoph Pucher) and C.P. (Claudio Poletti); visualization, H.V., K.W.N., C.P. (Christoph Pucher) and C.P. (Claudio Poletti); supervision, H.V.; project administration, K.W.N.; and funding acquisition, K.W.N. and H.V. All authors have read and agreed to the published version of the manuscript.

**Funding:** This research was funded by the Commission for Development Research (KEF) Austria grant number KEF P211 and the APC was funded by the OA publishing fund at BOKU library services.

**Institutional Review Board Statement:** Ethical review and approval were waived for this study due to the involvement of three Kenya government institutions the Kenya Forest Service (KFS), Kenya Wildlife Service (KWS) and Kenya Forest Research Institute (KEFRI) that provided us with the permission to conduct the research at Mt. Kenya, and their support with staff and records during data collection. The Mount Kenya Community Forest Associations leadership that work closely with KFS and KWS provided us with permission to involve their members actively in interviews and focus group discussions during data collection and as beneficiaries of this research project deliverables.

**Informed Consent Statement:** Informed consent was obtained from all subjects involved in the study.

**Data Availability Statement:** All used data sets in this research can be obtained from the online links that have been provided in Table 1 (Appendix A) that shows the data, description, source, when it was acquired, native resolution, extent, and processing.

**Acknowledgments:** We acknowledge the funds of the Commission for Development Research (KEF P211) and the APPEAR scholarship programme for providing us with financial support for the research. We also acknowledge the NASA, USA for providing us with satellite data of NDVI, fire occurrences and burnt area in Mount Kenya Forest Reserve and National Park from 2003 to 2018. We thank the management of BOKU for providing us with staff, office space, internet, printing and library services during the research period. We also thank the Kenya Forest Service, Kenya Wildlife Service and Kenya Forest Research Institute for providing us with the permission to conduct the research at Mt. Kenya, and their support with staff and records during data collection. We also acknowledge all the Mount Kenya Community Forest Associations for actively participating in interviews and focus group discussions during data collection. All individuals in the above mentioned institutions and associations have consented to this acknowledgement.

**Conflicts of Interest:** The authors declare no conflict of interest. The founding sponsors had no role in the design of the study; in the collection, analyses or interpretation of data; in the writing of the manuscript and in the decision to publish the results.

## Appendix A

**Table 1.** Used data sets.

| Data | Description | Source | Acquired | Native Resolution | | Extent | | Processing |
|---|---|---|---|---|---|---|---|---|
| | | | | Spatial | Temporal | Spatial | Temporal | |
| NDVI Boku (IVFL) | MODIS Products MOD13Q1 and MYD13Q1, Vegetation Indices 16-Day L3 Global 250 m SIN Grid, NDVI | http://ivfl-info.boku.ac.at/satellite-data-processing/dataprocess-global | 18 October 2020 | ~0.002° (~250 m), WGS84 | 7-day composites | 36.614, 38.099, -0.690, 0.5216 | Jan 2001 to Dez 2019 | Clip to study area, resample to CHIRPS data (same layout and resolution), bilinear interpolation |
| CHIRPS Precipitation | Rainfall Estimates from Rain Gauge and Satellite Observations | chg-ftpout.geog.ucsb.edu/pub/org/chg/products/CHIRPS-2.0/africa_monthly/tifs | 13 August 2020 | 0.05° (~5.5 km at the Equator), WGS84 | monthly | Africa | Jan 1981 to Dez 2019 | Clip to study area |
| MODIS Active Fire Data | MCD14DL Thermal Anomalies/Fire locations 1km FIRMS V006 NRT | https://firms.modaps.eosdis.nasa.gov/download/ | 14 August 2020 | 1 km (point represents center), WGS84 | daily | 36.85, 37.85, -0.5; 0.25 | Jan 2003 to Dez 2018 | None |
| MODIS Burned Area | MCD64A1 MODIS/Terra+Aqua Burned Area Monthly L3 Global 500 m SIN Grid | https://lpdaac.usgs.gov/tools/appeears/ | 17 August 2020 | ~0.0041° (~500 m), WGS84 | monthly | 36.75, 37.95, -0.6, 0.358 | Jan 2003 to Dez 2018 | Clip to study area |

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
