# Peer review of "Evaluation of the Relationship between Spatio-Temporal Variability of Vegetation Condition Index (VCI), Fire Occurrence and Burnt Area in Mount Kenya Forest Reserve and National Park"

_fire, doi:10.3390/fire6080282_

Round 1

Reviewer 1 Report

No suggestions.

Author Response

Dear Reviewer,

Please find attached our revised manuscript entitled ‘Evaluation of the relationship between spatio-temporal variability of vegetation condition index (VCI), fire occurrence and burnt area in Mount Kenya Forest Reserve and National Park’ to be considered for publication following your invitation that we received on the 16th May, 2023 to contribute a paper to your MDPI- Open Access Journal of Fire under the special issue: Understanding Heterogeneity in Wildland Fuels.

In this revised paper, we provide an unpublished overview of the relationship between VCI, occurrence of fires and burnt area in three land cover types: National Park (NP), Forest Stations (FS) and Farmlands (FL) in Mount Kenya Forest Reserve and National Park (MKFRNP). In this study, MODIS satellite data are used to derive the Normalized Difference Vegetation Index (NDVI) and the Vegetation Condition Index (VCI). The relationship between VCI and number of fires and burnt area in MKFRNP was analyzed using Poisson regression.

We have addressed all comments that were raised by the reviewers. We believe that the readers of the Journal will be informed how satellite data can be used to monitor the changing climate (meteorological factors), VCI, fire occurrence and burnt area and how our approach can contribute to the improvement of land use-specific fire monitoring, wildfire management strategies and habitat-specific firefighting in MKFRNP. The study is based on satellite data collected from 2003-2018, suggestions on data collection procedures are discussed as well.

The manuscript has not been published or simultaneously submitted for publication elsewhere. All authors have approved the manuscript and agree with the revisions that have currently been done for it to be submitted to the Fire Journal.

We look forward to hearing from you at your earliest convenience. Sincerely yours,

Dr. Kevin Wafula Nyongesa, 

Reviewer 2 Report

1. It is unusual for Africa to have both ground fires and crown fires as vegetation types and landscape there do not support those two types of fires.  Please kindly provide more information on that, why Kenya has both fire types as well?  

2. Understand that you use only MODIS data, but do you know about VIIRS on-broad Suomi NPP, NOAA-20, and NOAA-21?  They provide better active fire products through NASA-FIRMS (https://firms.modaps.eosdis.nasa.gov/) and NOAA-NESDIS (1. https://registry.opendata.aws/noaa-jpss/  and 2. https://noaa-jpss.s3.amazonaws.com/index.html), kindly check those out and perhaps add them in.  

3. Figure 1 needs to be improved to include the location of MKFRNP in Kenya and Africa, so the readers can get an idea of where it is located in Africa. 

4. Can you provide a vegetation cover map of MKFRNP?

5. Do you understand that a confidence percentage to MODIS active fire depends on the temperature values detected, right?  So it might mean that 0% detection might occur in the forested areas where the canopy might block the heat signal (https://www.earthdata.nasa.gov/faq/firms-faq#:~:text=For%20MODIS%2C%20the%20confidence%20value,pixels%20within%20the%20fire%20mask).  You need to be very careful of using such confidence % information. 

6. Add "The Climate Hazards Group InfraRed Precipitation with Station data (CHIRPS) is a quasi-global rainfall data set." at "CHIRPS," so readers know what it stands for.  Kindly check others as well.

7. Farmlands (FL) or Forest Stations (FS), you need to be consistence about those.  Kindly go through your texts again and change them accordingly.  

8. You have very good conclusions.

Need to check all abbreviations to make sure that they are correct and have what they all stand for.  English is okay.  

Author Response

Dear Reviewer,

We have responded to all your observations, questions and suggestions. 

 Please find attached our revised manuscript entitled ‘Evaluation of the relationship between spatio-temporal variability of vegetation condition index (VCI), fire occurrence and burnt area in Mount Kenya Forest Reserve and National Park’ to be considered for publication following your invitation that we received on the 16th May, 2023 to contribute a paper to your MDPI- Open Access Journal of Fire under the special issue: Understanding Heterogeneity in Wildland Fuels.

In this revised paper, we provide an unpublished overview of the relationship between VCI, occurrence of fires and burnt area in three land cover types: National Park (NP), Forest Stations (FS) and Farmlands (FL) in Mount Kenya Forest Reserve and National Park (MKFRNP). In this study, MODIS satellite data are used to derive the Normalized Difference Vegetation Index (NDVI) and the Vegetation Condition Index (VCI). The relationship between VCI and number of fires and burnt area in MKFRNP was analyzed using Poisson regression.

We have addressed all comments that were raised by the reviewers. We believe that the readers of the Journal will be informed how satellite data can be used to monitor the changing climate (meteorological factors), VCI, fire occurrence and burnt area and how our approach can contribute to the improvement of land use-specific fire monitoring, wildfire management strategies and habitat-specific firefighting in MKFRNP. The study is based on satellite data collected from 2003-2018, suggestions on data collection procedures are discussed as well.

The manuscript has not been published or simultaneously submitted for publication elsewhere. All authors have approved the manuscript and agree with the revisions that have currently been done for it to be submitted to the Fire Journal.

We look forward to hearing from you at your earliest convenience. Sincerely yours,

Dr. Kevin Wafula Nyongesa, 

Reviewer 3 Report

* What are the limitations/recommendations?

* Introduction: very good section, however, authors should need to update this section with more recent examples and I suggest developing the Introduction chap. also based on some projects results/official reports.
* The research questions should be more developed.

* What does it add to the subject area compared with other published material?

* Make sure to discuss your Conclusions in relation to other international studies hypotheses.

*The connection between results and some research projects could also be approached this could lead to an interesting and policy-relevant discussion. The discussion section should be improved by citing further references from international researchers.

Overall, the manuscript displays an original work.

Minor editing of English language required

Author Response

Dear reviewer,

We have tried and addressed all observations, suggestions and questions that had raised. 

Please find attached our revised manuscript entitled ‘Evaluation of the relationship between spatio-temporal variability of vegetation condition index (VCI), fire occurrence and burnt area in Mount Kenya Forest Reserve and National Park’ to be considered for publication following your invitation that we received on the 16th May, 2023 to contribute a paper to your MDPI- Open Access Journal of Fire under the special issue: Understanding Heterogeneity in Wildland Fuels.

In this revised paper, we provide an unpublished overview of the relationship between VCI, occurrence of fires and burnt area in three land cover types: National Park (NP), Forest Stations (FS) and Farmlands (FL) in Mount Kenya Forest Reserve and National Park (MKFRNP). In this study, MODIS satellite data are used to derive the Normalized Difference Vegetation Index (NDVI) and the Vegetation Condition Index (VCI). The relationship between VCI and number of fires and burnt area in MKFRNP was analyzed using Poisson regression.

We have addressed all comments that were raised by the reviewers. We believe that the readers of the Journal will be informed how satellite data can be used to monitor the changing climate (meteorological factors), VCI, fire occurrence and burnt area and how our approach can contribute to the improvement of land use-specific fire monitoring, wildfire management strategies and habitat-specific firefighting in MKFRNP. The study is based on satellite data collected from 2003-2018, suggestions on data collection procedures are discussed as well.

The manuscript has not been published or simultaneously submitted for publication elsewhere. All authors have approved the manuscript and agree with the revisions that have currently been done for it to be submitted to the Fire Journal.

We look forward to hearing from you at your earliest convenience. Sincerely yours,

Dr. Kevin Wafula Nyongesa, 

Round 2

Reviewer 3 Report

 Accept in present form